# Yeti claws: Cheliped sexual dimorphism and symmetry in deep-sea yeti crabs (Kiwaidae)

Christopher Nicolai Roterman[ID]1*, Molly McArthur[ID]1, Cecilia Laverty Baralle1, Leigh Marsh2, Jon T. Copley2

1 Institute of Marine Science, University of Portsmouth, Portsmouth, United Kingdom, 2 Ocean and Earth Science, University of Southampton, Waterfront Campus, Southampton, United Kingdom

* nicolai.roterman@port.ac.uk

**Data Availability Statement:** All relevant data are within the manuscript and its Supporting Information files.

## Abstract

Yeti crabs (Kiwaidae) are deep-sea hydrothermal vent and methane seep dwelling crustaceans that farm chemosynthetic microbes on their bodies. Sexual dimorphism is a common feature of decapod crustaceans, but little is known about its prevalence in species from deep-sea habitats. We address this knowledge deficit by investigating claw sexual dimorphism and symmetry in the hydrothermal-vent endemic 'Hoff crab', *Kiwa tyleri*. A total of 135 specimens from the East Scotia Ridge were examined, revealing mean asymmetry indices close to zero with respect to propodus length and height, albeit with a significantly larger number of marginally left-dominant individuals with respect to propodus length, possibly indicative of some task specialisation between claws, or a vestigial ancestral trait. Both male and female claws exhibit positive allometry with increasing carapace length, but males possess significantly larger claws compared with females when accounting for carapace size, exhibiting faster growing propodus length, and broader propodus heights throughout the size distribution. This marked difference is indicative of either male-male competition for mate access, sexual selection, or differential energy allocation (growth vs reproduction) between males and females, as observed in other decapod crustaceans. In contrast, a reanalysis of data for the methane seep inhabiting yeti crab *Kiwa puravida* revealed no significant difference in claw allometry, indicating a possible lack of similar sexual selection pressures, and highlighting potential key differences in the ecological and reproductive strategies of *K. tyleri* and *K. puravida* relating to claw function, microbial productivity and population density. Whether sex differences in claw allometry represents the norm or the exception in Kiwaidae will require the examination of other species in the family. This research enhances our understanding of the behaviour, ecology and evolution of yeti crabs, providing a basis for future studies.

## Introduction

In contrast to the general pattern of photosynthetically derived food limitation in the deep sea, hydrothermal vents and hydrocarbon (methane) seeps host high biomass oases of endemic macrofauna sustained by the primary productivity of chemosynthetic microbes [1]. These

**Funding:** Specimens collection for this study was funded by the Natural Environment Research Council in the form of a grant (NE/D01249X/1) received by JTC. No additional external funding was received for this study.

**Competing interests:** The authors have declared that no competing interests exist.

communities, first discovered in 1977 [2] experience steep environmental gradients in the mixing zone between ambient seawater and fluids emanating from the seafloor, which are typically anoxic, acidic, and in the case of hydrothermal vents very hot (temperatures > 350°C) and rich in dissolved metals and other compounds [1]. Consequently, macrofauna recovered from these habitats exhibit a variety of physiological adaptations to tolerate hypoxia [3, 4], thermal stress, protection and detection [5–12], and toxicity [13, 14]. Strikingly, these animals have often developed intimate symbioses with chemosynthetic microbes [15], either farming microbes on modified appendages [16–18] or internally [19–22], ensuring a reliable and plentiful food source. However, owing to the remoteness of such habitats, the challenges inherent in their sampling and long-term monitoring, and difficulties in maintaining specimens in aquaria, little is known about the behaviour, ecology and life-history of these animals compared with shallow-water species.

One such example are Kiwaidae, also known as yeti crabs: a decapod crustacean family within the infraorder Anomura, first discovered in 2005 [23] and found exclusively at deep-sea chemosynthetic ecosystems, comprising five described species within the genus *Kiwa* Mac-Pherson *et al.* 2005 [23–27] and one undescribed [28] from the Pacific, Southern and Indian Oceans. These squat lobsters likely radiated within the last 45 million years [28, 29] and appear to be sustained primarily by farming chemosynthetic microbes on hair-like setae extensively covering their exoskeleton (hence their common name), as indicated by the morphological examination of setae [27, 30, 31] feeding appendages [24–27], and isotope analyses [32–34]. All species appear to lack eyes, [23–27] and the basal bauplan for the family is individuals exhibiting disproportionately long, slender chelipeds with plumose or bristly setae [28] (Fig 1). An exception to this is *Kiwa tyleri* Thatje *et al.* 2015 from the East Scotia Ridge in the Southern Ocean, commonly known as the 'Hoff crab', and also a closely-related undescribed species *Kiwa* sp. from the Southwest Indian Ridge, herein referred to as *Kiwa* sp. SWIR, which have a more compact overall shape, shorter chelipeds and a greater profusion of setae on their ventral surface than on the chelipeds [28, 35]. *K. tyleri* was observed in exceptionally high densities (> 1000 individuals m$^{-2}$) at the base of vent chimneys, with larger individuals up the sides of the chimneys [36, 37], living within a thermal envelope bounded by ambient sea temperatures ranging from 0°C to -1.3°C [36], which are typically too cold for crab and lobster survival [38–40]. It has been hypothesised, therefore, that the distinctive body shape and placement of setae in this species is an adaptation to limited space and the vertical terrain on which *K. tyleri* must survive, constrained within a thermal bubble [25, 41]. Yeti crabs produce relatively few but large lecithotrophic larvae, exhibiting extremely abbreviated development and hatching as megalopa following a likely prolonged brooding period [41, 42]. Their mode of dispersal is thereby less optimised for long-range dispersal than local retention on patchy habitats, which may partly explain the extreme isolation of some species [28].

At present, the longevity of kiwaid individuals, the age when they become reproductively active and their lifelong fecundity are unknown, but some aspects of reproductive behaviour have been inferred from spatial analyses. Spatial mapping and sampling of *K. tyleri* at vents on the East Scotia Ridge in the Southern Ocean indicate the largest individuals, which are found closest to the hot vent effluent, are males, with smaller, brooding females found more in the vent periphery, possibly trading access to microbial food production for more oxygenated conditions in which to brood their eggs [41, 43]. While this spatial pattern has only been observed in *K. tyleri*, an extensive morphological study (> 200 specimens) of the tropical East Pacific methane seep-associated *Kiwa puravida* Thurber *et al.* 2011 by Azofeifa-Solano *et al.* [44] found the largest individuals to be males, as did the recent species description of *Kiwa gemma* Liu *et al.* 2024 from vents on the Galapagos microplate [27], possibly indicating a similar spatial pattern occurs with those species. This apparent sexual dimorphism may be suggestive of

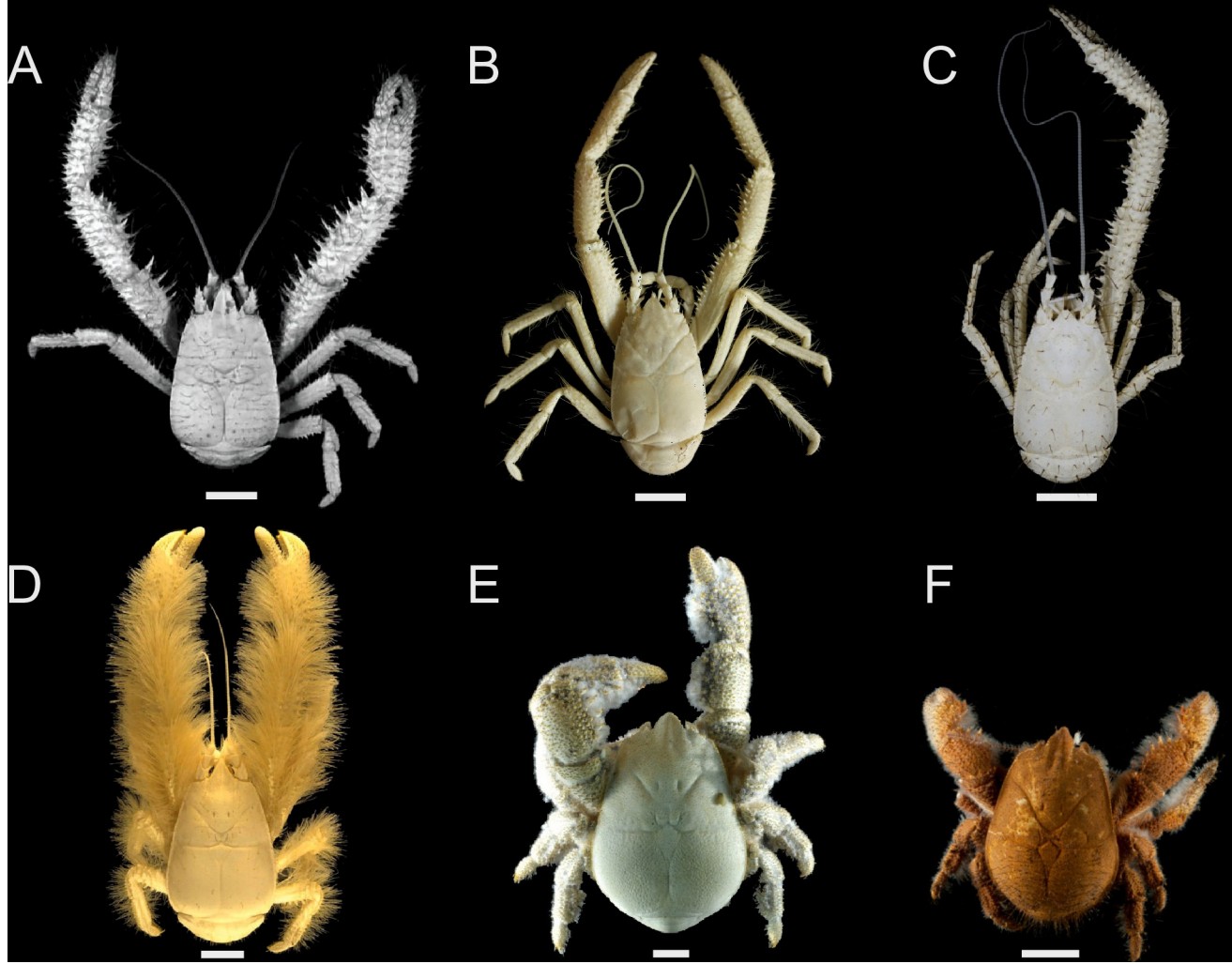

**Fig 1. Dorsal view of the known yeti crab (Kiwaidae) species, from Roterman et al. [28].** Scale bars are approximate and represent 10 mm. A) *Kiwa puravida*, B) *Kiwa gemma*, C) *Kiwa araonae*, D) *Kiwa hirsuta*, E) *Kiwa tyleri*, F) *Kiwa* sp. SWIR.

sexual selection [45] or at least differential growth investment as a consequence of the environmental constraints that males and females experience as a function of their reproductive ecology [41].

Chelipeds are the first pair of pereopods in decapod crustaceans with the final two segments comprising the claw (chela) with a movable finger (dactylus) and a larger fixed segment (propodus). For clarity, we use the term cheliped in reference to the whole appendage, and claw to the final two segments of the appendage (propodus and dactylus) throughout this manuscript. These have a variety of uses, including as weapons of offense and defence, for predation, collecting suspended food particles, intraspecific competition for mates, visual display and generating sound [46]. A common, but not exclusive feature amongst decapods is heterochely, where one cheliped/claw is larger or differently shaped to the other and is related to task specialisation relating to durophagy, such as having a crushing and a cutting claw [47, 48], but may also in some species be sexually dimorphic, reflecting male-male competition, differential energy allocation between sexes, or sexually selection [46, 49]. Within Kiwaidae, the chelipeds appear to be primarily used for farming microbes, based on their dense covering of setae, with

the possible exception of *K. tyleri* and *Kiwa* sp. SWIR, as already mentioned. Thurber et al. [24] reported *K. puravida* actively waving their chelipeds, presumably to enhance microbial production, although this behaviour has not been observed in the hydrothermal vent species. The claws of *K. puravida*, *Kiwa araonae* Lee *et al.* 2016 and *K. gemma* which belong to the 'bristly' clade [28] are similar in being slender, with a prominent tooth on the cutting edge of the dactylus and row of teeth on the cutting edge of the propodus, suggesting microbial production is not the only function. Only one specimen of *Kiwa hirsuta* MacPherson *et al.* 2005 (a male), was collected and its claws–while stouter and broader than the afore mentioned species–appear to have similar, if not slightly less prominent, dentition. However, teeth appear to be absent on *K. tyleri* claws. The presence of claw dentition may suggest some kiwaid diets are not exclusively microbial and *K. hirsuta* individuals were observed eating bathymodiolid mussels, which had been damaged by the submersible [23], but it is unclear if they would have been capable of breaking into the mussels without assistance. The description of *K. gemma* notes that the female claws are more slender with less prominent teeth than the males, possibly indicative of intraspecific competition between males [27].

As well as possible differences in claw dentition between the sexes, the overall size and shape of the chelipeds between the sexes have been previously remarked upon. The large individuals of *K. tyleri* collected from close to the vent orifice, all of which were males (Fig 2), were described as having longer and broader chelipeds [25, 41, 43] compared with smaller individuals collected further away, and interactions described as 'fighting' recorded between pairs of large individuals [43]. Liu *et al.* [27] observed that amongst the 18 examined specimens of *K. gemma*, the typically larger male specimens exhibited chelipeds around three times the length of the carapace, versus 1.6 times in females. To date only one study had attempted to explicitly explore claw sexual dimorphism in Kiwaidae. Azofeifa-Solano *et al.* [44] measured more than 200 *K. puravida* specimens comparing the length and breadth of the claw propodus vs carapace length, revealing the individuals with the largest claws in proportion to the carapace to be male, which were also the largest individuals. Their analysis, however, assumed isometry with respect to claw and carapace growth, which was found herein to not be the case, therefore precluding any inferences regarding sexual dimorphism when accounting for size.

This study aims to gain further insight into the function of kiwaid chelipeds by exploring the extent of claw asymmetry in specimens of *K. tyleri*, and to determine whether males have larger claws than females when accounting for size, indicative of differential energy allocation between males and females due to natural or sexual selection. For comparative purposes *K. puravida* measurements from Azofeifa-Solano *et al.* [44] have also been re-analysed to determine any general patterns within Kiwaidae.

## Materials & methods

Specimens of *K. tyleri* were collected from hydrothermal vents on the E2 and E9 ridge segments of the East Scotia Ridge [25, 36] in the Scotia Sea–part of the Atlantic sector of the Southern Ocean–on two separate NERC-funded expeditions aboard the *RRS James Cook* (JC042–07 January 2010 to 24 February 2010; JC080–02 December 2012 to 30 December 2012), as part of the ChEsSO project. The E2 vent field is situated between 56˚05.29' and 56˚05.49'S and 30˚19.00' and 30˚19.36'W at ~2600 m depth and the E9 vent field is ~450 km further South at ~2400 m depth, between 60˚02.50' and 60˚03.00'S and 29˚59.00' and 29˚58.60'W. Ambient sea temperatures at the periphery of the vent fields ranged from 0˚C to -1.3˚C at E2 and E9 respectively, and maximum fluid temperatures emanating from black-smoker chimneys were recorded as 352.6 ˚C and 382.8 ˚C at E2 and E9 respectively [36]. Kiwaids at the base of chimneys were typically in temperatures ranging from 2–12˚C, although

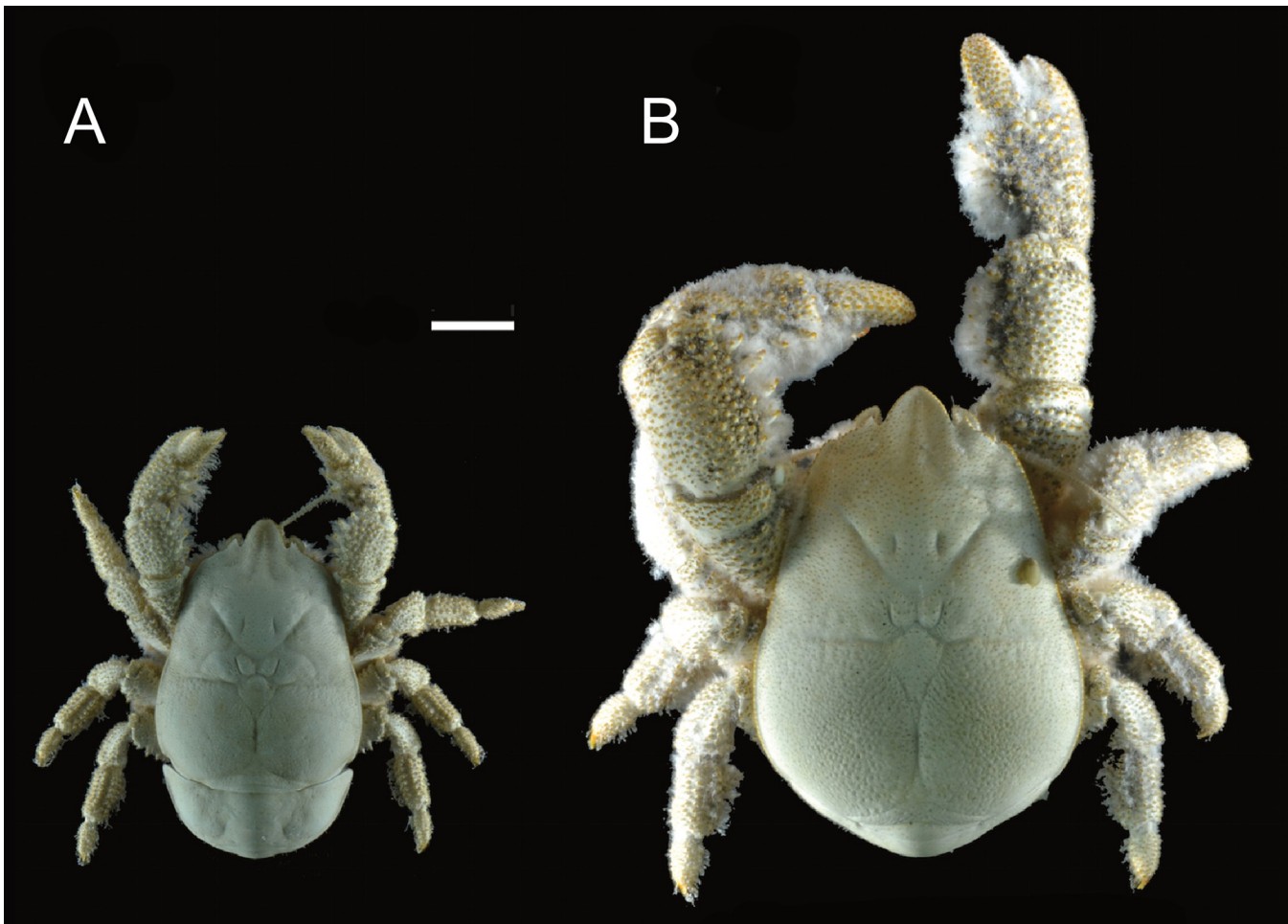

**Fig 2.** Female (A) and male (B) *Kiwa tyleri* type specimens, modified from Thatje *et al.* [25]. Scale bar is approximate and represents 10 mm.

individuals on the chimney sides may have experienced higher temperatures in closer proximity to the vent orifice [41, 43]. Individuals were sampled from areas of pillow basalt with diffuse venting as well as from the base and sides of 'black smoker' chimneys with a suction sampler deployed from the ROV *ISIS* on dives 130, 132, 134, 135, 189 (E2) and dives 140, 141 and 144 (E9). Kiwaids were not randomly sampled and different locations had distinct size cohorts with skewed sex ratios [37, 41, 43]. Aboard *RRS James Cook*, recovered specimens were triaged in a 4°C temperature-controlled room and either preserved initially in formalin (5% formaldehyde and buffered seawater) before being preserved in 70% ethanol, or preserved in 99% molecular grade ethanol, or frozen at -80°C for subsequent analyses ashore.

*Kiwa tyleri* individuals were sexed depending on the presence of gonopores on the first segment of the third pereopod, which is only exhibited in females [25]. Structures were measured using digital vernier callipers to an accuracy of 0.1 mm. Carapace length was measured from the base of the rostrum (defined as where the supraocular spine meets the rostrum) to the posterior dorsal margin of the carapace. The length (from proximal to distal margins), and height (maximum distance across the segment at a right angle to the length) of the propodus of both right and left chelipeds was measured (Fig 3) to characterise the relative growth of claws, and for direct comparison with the data published by Azofeifa-Solano *et al.* [44].

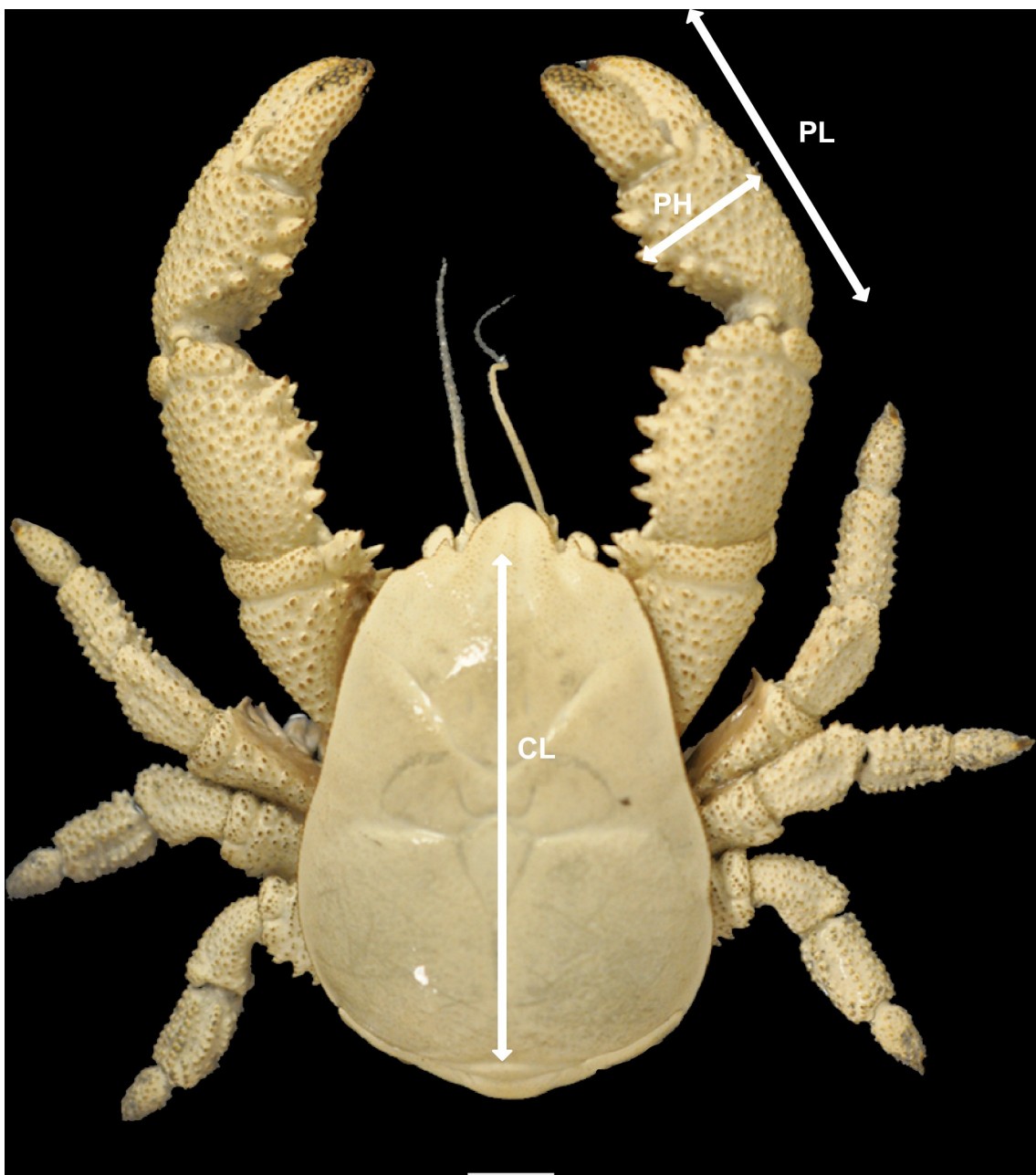

**Fig 3. Measurements of the external morphology of *Kiwa tyleri*.** CL = carapace length, PL = cheliped propodus length, PH = cheliped propodus height. Scale bar = 10 mm (1 cm).

Propodus length and height symmetry in individuals with both chelipeds was assessed by calculating an asymmetry index (AI) according to Van Valen [50]; AI = (Right—Left) / (Right + Left). Negative values indicate a left skew and positive values a right skew. Chi-square tests were performed to see if the proportion of left-dominant or right-dominant individuals deviated from a 1:1 ratio with respect to propodus length and height. After data were determined not to be normally distributed or exhibit equal variances, deviations in propodus length and height AI from zero were tested with the non-parametric Wilcoxon Signed Rank test and

differences between males and females were assessed with Wilcoxon Rank Sum tests. For all subsequent body proportion analyses, the larger or dominant claw measurement was used. To detect differences in allometric scaling (carapace vs propodus relative growth rate) between male and female *K. tyleri* specimens, measurements were log-transformed, given the tendency for morphological structures such as claws to scale allometrically (non-linearly) relative to other structures [51–53]. As bivariate relative growth is being investigated where both variables are subject to error, standardised (reduced) major axis (SMA) regressions (type II regressions) were performed using the smatr R package [54, 55]. Homoscedasticity and normality of residuals was assessed with residuals vs fitted values plots, and Q-Q plots. To test for common slope between males and females, the smatr package employed a likelihood ratio test and where slopes were not found to be significantly different, slope elevation (y-axis offset) was compared using the Wald statistic between regression models assuming equal slope [55].

In addition to the *K. tyleri* data acquired herein, published data from Azofeifa-Solano *et al.* [44] on *K. puravida* carapace length vs cheliped propodus length and height (reported as width by the authors) were re-analysed according to the above methodology for comparative purposes. All statistical analyses and data visualisation was performed in R studio [56].

## Results

A total of 135 *K. tyleri* specimens were measured, of which 109 had both chelipeds fully intact. 67 (49.6%) individuals measured were female and 68 (50.4%) were male. Carapace lengths ranged from 15.2 mm to 52.7 mm in females (mean = 29.4 mm, ± 9.37 mm standard deviation) and from 13.7 to 72.5 mm in males (mean = 38.6 mm, ± 14.32 mm standard deviation). Raw measurements are reported in S1 Table and mean appendage measurements are reported in Table 1.

Of the 109 specimens with two chelipeds, 17 exhibited no measurable deviation in claw symmetry (AI = 0) for propodus length, with 54 left dominant and 34 right dominant. 21 exhibited no symmetry deviations for propodus height, with 38 left dominant and 49 right dominant. Mean AI values were close to 0, with -0.0057 (± 0.024 standard deviation) for propodus length and -0.00059 (± 0.0268 standard deviation) for propodus height (Fig 4). The maximum deviation from symmetry for length was 0.181 i.e., where the left propodus was 44.3% longer than the right propodus. The maximum symmetry deviation for propodus height was 0.153, where the left propodus was 36% broader than the right propodus. Both these extreme asymmetries were found in the same individual. The median size discrepancy between claws was 1.2% for propodus length, 1.7% for height with 4 individuals having a discrepancy greater than 10% for length, and 6 having a discrepancy greater than 10% for height. Wilcoxon Rank Sum tests revealed no significant difference in AI between males and females for propodus length (W = 1346, p = 0.410) and height (W = 1269, p = 0.250) and the Wilcoxon Sign Rank test found a significant AI deviation from zero (V = 1523, p = 0.035), but not for height

**Table 1. Summary of *Kiwa tyleri* measurements including *Kiwa puravida* measurements extracted from Azofeifa-Solano *et al.* [44].** Mean measurements are shown with ± standard deviation. N = number, CL = carapace length, PL = cheliped propodus length, PH = cheliped propodus height.

| Species | Sex | N | CL (mm) | PL (mm) | PH (mm) |
|---|---|---|---|---|---|
| *Kiwa tyleri* | Female | 67 | 29.4 ± 9.37 | 13.9 ± 5.08 | 8.1 ± 2.98 |
| | Male | 68 | 38.6 ± 14.32 | 21.7 ± 10.69 | 13.2 ± 5.98 |
| | Total | 135 | 34.1 ± 12.93 | 17.8 ± 9.22 | 10.7 ± 5.39 |
| *Kiwa puravida* | Female | 68 | 8 ± 6.12 | 5.3 ± 4.65 | 1.8 ± 1.48 |
| | Male | 138 | 6.6 ± 7.32 | 4.6 ± 6.30 | 1.5 ± 1.98 |
| | Total | 206 | 7 ± 6.97 | 4.8 ± 5.81 | 1.6 ± 1.83 |

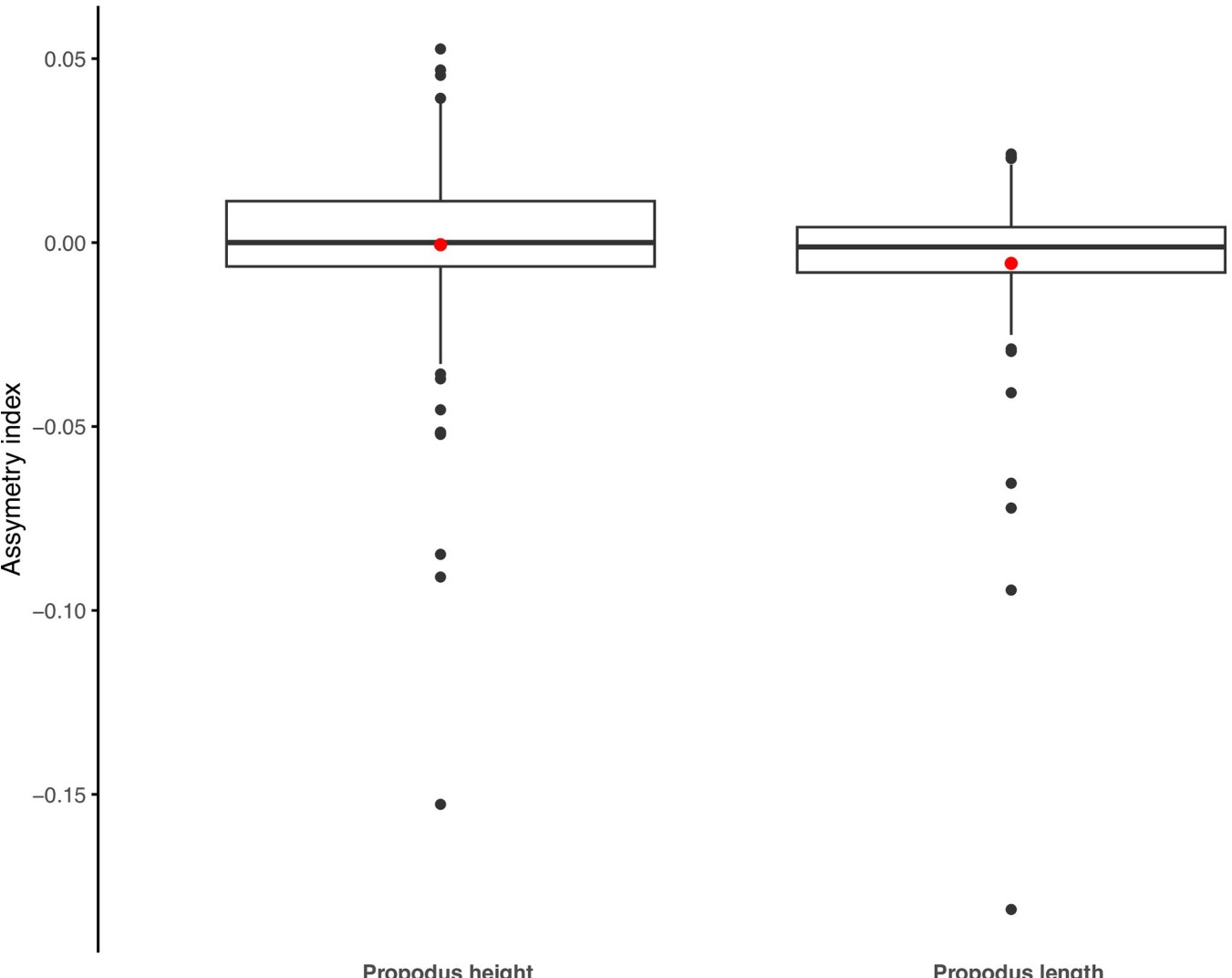

**Fig 4. Box and whisker plot showing asymmetry index (AI) values for cheliped propodus length and height in 109 specimens of *Kiwa tyleri*.** Red dot = mean; thick horizonal line = median. The box represents the interquartile range (middle 50% of the data) and whiskers extend 1.5 * interquartile range from the lower and upper quartiles, respectively. Black dots represent outliers (outside 1.5 * interquartile range).

(V = 2184.5, p = 0.253). Likewise, chi-square revealed a significant deviation from 1:1 left-right dominance with respect to propodus length (chi-square = 4.546, df = 1, p = 0.033) with significantly more left dominant individuals, but not for height (chi-square = 1.391, df = 1, p = 0.238).

Cheliped propodus lengths and heights for *K. tyleri* did not scale linearly with carapace length (Fig 5). In all cases, the relative size of the claw dimension increased in relation to carapace size as it increased (positive allometry), as is the case with the *K. puravida* cheliped propodus dimensions taken from Azofeifa-Solano *et al*. [44] (Fig 6). All SMA regression models explained a high proportion of the data variability ($r^2 > 0.96$; $p < 2.22 \times 10^{-16}$, see S2 Table). For *K. tyleri*, there was a significant difference in slope between log carapace length on log propodus length between males and females ($p = 9.47 \times 10^{-7}$) with a steeper slope for males (Fig 5), i.e. male propodus lengths becoming proportionately larger than females with increasing carapace length, indicative of faster claw growth rates. For log propodus height, there was no

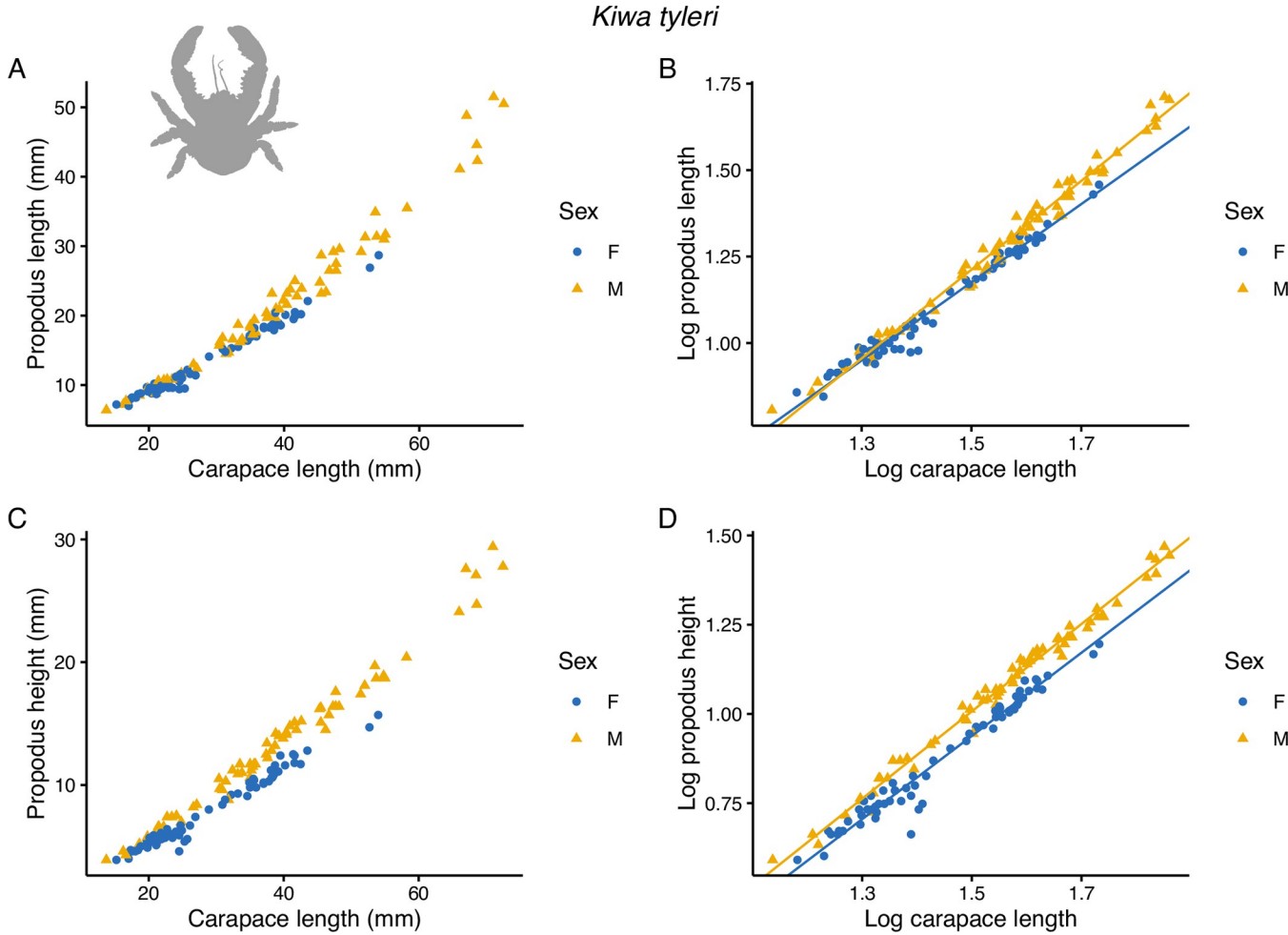

**Fig 5. Scatter plots generated in Rstudio of dominant cheliped propodus dimensions for 135 specimens of *Kiwa tyleri* collected from from hydrothermal vents on the East Scotia Ridge in the Southern ocean.** A) propodus length vs carapace length and B) log-transformed data. C) propodus height vs carapace length and D) log-transformed data. Blue dots are females and orange triangles are males. Lines in B) and D) are SMA regressions fitted to the data calculated in the smatr R package.

significant difference in the angle of slope between males and females (p = 0.078), but there was a highly significant difference in slope elevation when slope angles were assumed to be equal (p < 2.22 x 10$^{-16}$) (Table 2), with males having broader log propodus heights than females throughout the size distribution of log carapace lengths (Fig 5). Given the differences in positive allometry with respect to claw lengths but not heights, as males grow in size their claws become more slender in shape at the upper end of the size spectrum. Conversely, comparisons between male and female SMA regression models for *K. puravida* from Azofeifa-Solano *et al.* [44] revealed no effect of sex on slope angle for log propodus length (p = 0.951) or height (p = 0.414) vs log carapace length, and no effect of sex on slope elevation when assuming equal slope angles (log propodus length, p = 0.277; log propodus height, p = 0.537) (Fig 6, Table 2) indicating no differences in claw allometry between males and females.

## Discussion

This is the first exploration of claw symmetry in yeti crabs (Kiwaidae), indicating that *K. tyleri* exhibits a low, but significant degree of asymmetry in cheliped propodus length (but not

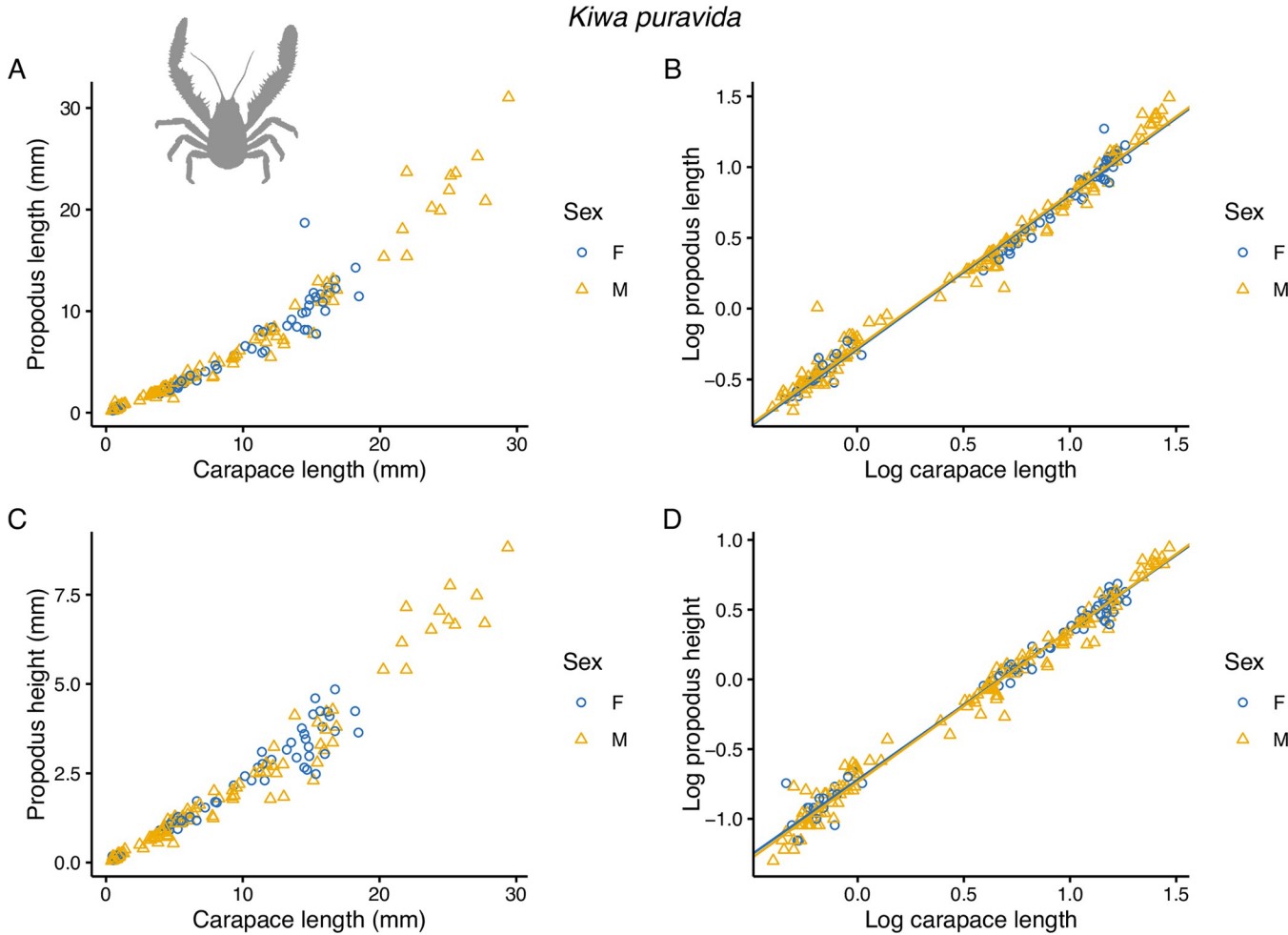

**Fig 6. Scatter plots of cheliped propodus dimensions for 206 specimens of *Kiwa puravida* taken from Azofeifa-Solano *et al.* [44].** A) propodus length vs carapace length and B) log-transformed data. C) propodus height vs carapace length and D) log-transformed data. Blue circles are females and orange hollow triangles are males. Lines in B) and D) are SMA regressions fitted to the data calculated in the smatr R package.

height), with asymmetry indices (AI) for most individuals close to zero and no significant differences in AI between sexes. While some tests for heterochely were significant ($0.01 < P < 0.05$), we acknowledge that the relatively small dataset of 109 individuals necessitates a cautious interpretation of the results, and further data collection will be necessary to confirm heterochely. While most individuals had claws of very similar size, one outlier individual whose left cheliped propodus was over 44% larger than the right, may reflect a developmental asymmetry, disease, parasitism, or the consequence of damage and/or cheliped loss and subsequent regrowth, as reported for some decapods [46]. There is no evidence of any of the specimens in this study having lost a cheliped prior to sampling, however, and loose chelipeds were found at the bottom of the containers in which specimens were batch-preserved, suggesting that they detached after specimen triage.

The very slight, but significant claw asymmetry in *K. tyleri*, with a tendency to left dominance in claw length (Fig 4) may provide insights into claw function in this species. Asymmetry (heterochely) in decapod claws is generally associated with specialisation relating to food handling/manipulation, sound generation, sexually-selected activity e.g. display in fiddler crabs, or conspecific fighting in (male and female) aeglid squat lobsters [46, 57–59]. However,

**Table 2. Standardised major axis (SMA) regression model output showing the relationship between log-transformed propodus length/height and carapace length for male and female *Kiwa tyleri* and *Kiwa puravida* specimens as performed by the smatr R package.** Models initially compared slope angle between males (M) and females (F) and if non-significant, a comparison with slope elevation (y-axis offset) assuming equal slopes was made. 95% CI = 95% confidence interval, H0 = null hypothesis of test comparing male and female slopes angle/elevation, P = p-value, indicating significance of test with values < 0.05 highlighted in bold. For complete model outputs, see S2 Table.

| Species | Model type | Comparison | Estimate | 95% CI | H0 | Statistic | P |
|---|---|---|---|---|---|---|---|
| *Kiwa tyleri* | log propodus length ~ log carapace length * sex | slope (F) | 1.130 | 1.088–1.175 | equal slopes | likelihood ratio 24.03 (1df) | **0.000** |
| | | slope (M) | 1.284 | 1.245–1.323 | | | |
| | log propodus height ~ log carapace length * sex | slope (F) | 1.165 | 1.112–1.222 | equal slopes | likelihood ratio 3.1 (1df) | 0.078 |
| | | slope (M) | 1.224 | 1.191–1.257 | | | |
| | log propodus height ~ log carapace length + sex | intercept (F) | -0.873 | -0.915 – -0.831 | equal elevation | Wald statistic 174.9 (1df) | **0.000** |
| | | intercept (M) | -0.806 | -0.851 – -0.761 | | | |
| *Kiwa puravida* | log propodus length ~ log carapace length * sex | slope (F) | 1.088 | 1.059–1.118 | equal slopes | likelihood ratio 0.004 (1df) | 0.951 |
| | | slope (M) | 1.087 | 1.063–1.112 | | | |
| | log propodus length ~ log carapace length + sex | intercept (F) | -0.287 | -0.308 – -0.266 | equal elevation | Wald statistic 1.179 (1df) | 0.277 |
| | | intercept (M) | -0.275 | -0.292 – -0.258 | | | |
| | log propodus height ~ log carapace length * sex | slope (F) | 1.074 | 1.041–1.107 | equal slopes | likelihood ratio 0.668 (df) | 0.414 |
| | | slope (M) | 1.091 | 1.067–1.115 | | | |
| | log propodus height ~ log carapace length + sex | intercept (F) | -0.727 | -0.750 – -0.704 | equal elevation | Wald statistic 0.381 (1df) | 0.537 |
| | | intercept (M) | -0.734 | -0.751 – -0.717 | | | |

there is continued debate regarding the links between ecology and heterochely [49]. The bythograeid crab *Austinograea williamsi* Hessler & Martin 1989, found at vents in the Mariana Back-Arc Basin often exhibits larger crusher and more slender 'cutter' claws commonly observed in other brachyuran predators of shelled organisms (durophagy) [47, 60]. The more subtle heterochely in *K. tyleri* may therefore be consistent with claws being used primarily for functions other than carnivory of hard or tough food sources, such as microbial food production, intraspecific agonistic interactions, or manipulation of more delicate food types. The primary reliance of *K. tyleri* on farming microbes for sustenance is indicated by *in situ* observation, morphology of feeding appendages and setae, and analyses of $\partial^{13}$C, $\partial^{15}$N and $\partial^{34}$S values indicating chemosynthetically-derived nutrition [32, 25, 31], but this doesn't preclude the possibility of carnivory of other vent-endemic fauna.

Spani *et al.* [49] found a strong phylogenetic signal with heterochely, which suggests that a low degree of claw asymmetry and a left-handed tendency may be characteristic of Kiwaidae, but examination of the other species within the family will be necessary to confirm this. To date, we are unaware of heterochely being investigated in the other two sister families, which, along with Kiwaidae comprise the superfamily Chirostyloidea: Chirostylidae and Eumunididae. The feeding ecology of these two deep-water coral-associated families is presently unclear, although some observations suggest they may use corals as platforms for collecting suspended food particles or catching pelagic mesofauna [61–63], rather than the durophagy often associated with Brachyura [46]. Both families exhibit prominently elongated chelipeds, similar to the yeti crab bauplan, and it may be that this mode of feeding in Chirostylidae and Eumunididae is ancestral for the superfamily based on phylogeny [26], even though the only-known stem-lineage kiwaid fossil was not found in association with coral deposits [64]. Although heterochely has not been investigated in Chirostyloidea, it has in Aegloidea, a superfamily basal to Chirostyloidea [28]. Aeglid squat lobsters, which inhabit freshwater habitats in South America exhibit consistent and prominent left claw dominance, which is most visually apparent in males but also present in females, and is associated with agonistic interactions related to territoriality, visual display and mate guarding [65, 66]. The absence of any significant difference in

heterochely between males and female *K. tyleri* specimens, might suggest that male-male agonistic interactions are not the primary cause for the heterochely in the same way as aeglid squat lobsters, but more specimens may need to be measured to confirm the lack of sex differences. In other Anomura, such as porcelain crabs (Porcellanidae), heterochely has been reported, either without a tendency for claw dominance of a particular side, or with right dominance and is similarly associated with territoriality, male-male competition and mate guarding [67]. In munidid squat lobsters (Munididae) isochely has been reported with occasional heterochely attributed to regeneration after autotomy [68]. Whether the slight heterochely and tendency to left claw dominance in *K. tyleri* reflects some degree of task specialisation relating to competition or food handling or is an ancestral trait within Chirostyloidea which is no longer functional warrants further investigation.

The results of the standardised major axis (SMA) regression models clearly indicate that *K. tyleri* males exhibited faster growth rates for cheliped propodus lengths relative to carapace length (positive allometry) and throughout the size distribution had generally broader and longer claws, than females (Fig 5). *Kiwa puravida* also exhibited positive claw allometry, but in contrast with *K. tyleri* there was no difference in positive allometry with respect to sex. The reanalyses of *K. puravida* data contradicts Azofeifa-Solano *et al.* [44], which can be attributed to the use of log transformation in this study, which was necessary given that lack of linear scaling between carapace length and claw dimensions (Figs 5 and 6).

This is the first quantitative morphological study to show differences in yeti crab claw allometry between males and females. The difference in cheliped size between males and females was previously observed in *K. tyleri* [36, 37, 41, 43], but the size difference between males and female specimens was not accounted for (Fig 2). This difference in claw growth may reflect differential energy investment between the two sexes, with females investing more in gamete production and larval brooding and males investing more in growth, driven by competition for optimal locations adjacent to hydrothermal vent effluent where microbial productivity is higher. Differential investment in growth vs reproduction has also been used to explain sexual dimorphism in galatheoid squat lobsters [69]. Additionally, females were also observed further away from the anoxic vent effluent than the males, possibly owing to the need to keep eggs and larval broods well oxygenated; thereby limiting access to microbial sustenance [41]. A food-limitation hypothesis for smaller female size is supported by the observation that brooding female *K. tyleri* individuals collected at the periphery of vents were typically rust-coloured, presumably as a result of oxidation of iron compounds deposited on the exoskeleton, indicative of a larger elapsed time since moulting compared with individuals collected closer to the vents [41].

As well as differential energy allocation, the differences in claw allometry between *K. tyleri* males and females may also be indicative of male-male competition for territory/mating opportunities, consistent with observations of individuals apparently 'fighting' [24, 43], or possibly sexual selection of larger claws by females, or a combination of the two. Larger claws in males vs similarly-sized females is widely reported in decapod crustaceans [46, 69, 70] and is typically attributed to either agonistic male-male interactions or visual display for access to females (sexual selection). Given the apparent absence of functioning eyes in Kiwaidae, the possibility of larger male claw size being sexually selected for by females would be contingent on physical contact rather than visual display, although evidence for this form of sexual selection is presently lacking. Asakura *et al.* [70] listed several mating systems in decapod crustaceans, but the 'precopulatory guarding' type where males resource guard a female for a period to time prior to mating, arising when male-male competition for females is intense, and/or the window for mating with a receptive female is time constrained, is more typical in anomuran squat lobsters [69]. In squat lobsters, the male typically embraces the female dorsally, holding

her in place by grasping with the cheliped or pereopods [69] and Thatje and Marsh [43] observed that *K. tyleri* males exhibited a substantially-enlarged "comb of strong, flattened, teeth-like spines" on the ischium to carpus cheliped segments which would allow the male to "arrest the female on its lateral margins of the carapace". If their interpretation is correct, this would suggest that *K. tyleri* conforms to the precopulatory mating type, as Azofeifa-Solano *et al.* [44] concluded with *K. puravida*, which would be consistent with male-male competition for access to mating opportunities.

The absence, however, of sex differences in claw allometry in *K. puravida* does warrant explanation. One possibility could relate to differences in size between *K. tyleri* and *K. puravida* in the study. Reviewing sexual dimorphism in squat lobsters, Thiel & Lovrich [69] noted that in larger species, males tend to become sexually mature at smaller sizes than females compared with smaller species where males and females reach maturity at similar sizes. *Kiwa tyleri* appears generally much larger than *K. puravida*, with the largest *K. puravida* individual reported by Azofeifa-Solano *et al.* [44] (carapace length = 29.4mm) being far smaller than the largest *K. tyleri* individual (72.5 mm) reported here. *Kiwa tyleri* males may reach sexual maturity at smaller sizes in their size distribution, exhibiting sexual dimorphism comparatively earlier. If *K. puravida* males reach sexual maturity only at the very upper end of their size distribution, then any detection of differences in positive allometry would hypothetically be reliant on measuring sufficient numbers of as-yet unsampled larger males exhibiting disproportionately larger claws. However, no such *K. puravida* individuals have been reported thus far [24, 44]. In contrast with *K. puravida*, even the smallest individuals of *K. tyleri* exhibited claw proportion differences between the sexes suggesting an ecological difference between the two species.

*Kiwa tyleri* and *Kiwa* sp. SWIR are notable amongst kiwaids in having the preponderance of their setae on their ventral surface, rather than on elongated chelipeds. It may be therefore that the use of the chelipeds as a principal surface for farming microbes in *K. puravida* (and possibly other elongated kiwaids–Fig 1) places an evolutionary constraint on the selection for oversized male claws in those taxa. Thurber *et al.* [24] reported *K. puravida* individuals waving their chelipeds back and forth in a rhythmic fashion, which is a behaviour not seen in other kiwaids. The inference is that *K. puravida*, inhabiting low-flow methane seeps wave their chelipeds to enhance fluid flow and boost microbial productivity, suggesting their chelipeds are critical for food production. In contrast, *K. tyleri* may be less reliant on chelipeds for sustenance, not only due to the location of their setae, but also if the vents they inhabit engender greater microbial production (as evidenced by their considerably larger maximum body size). A reduced reliance on chelipeds for microbial production would free them for other functions, such as weapons in male-male agonistic interactions, or as sexually selected indicators of fitness. *Kiwa tyleri* are also far more numerically dominant than *K. puravida*, often existing in very high densities [37, 71] and the difference in claw sexual dimorphism between the two species may reflect more intensive competition between *K. tyleri* individuals inhabiting a narrow viable thermal envelope around the Southern Ocean vents, as is shown for spatially constrained cave-dwelling munidopsid squat lobsters [72].

If the prevalence and intensity of male-male competition is similar between *K. puravida* and *K. tyleri*, it is also possible that *K. puravida* exhibits sexual dimorphism in cheliped segments as yet unmeasured, such as the enlarged comb-like spines on the proximal cheliped segments observed by Thatje and Marsh [43] in *K. tyleri*, and indeed, if all the spines on the chelipeds and/or the teeth on the cutting edge of the claw are enlarged. Calverie and Smith [68] observed male munidid squat lobsters exhibiting more prominent teeth on the claw dactylus (finger) suggesting their use for puncturing the chelipeds of male rivals and preliminary observations hint this may be the case with *K. gemma*, which belongs to the same bristly clade

as *K. puravida* [27]. It is notable in this context that *K. tyleri* appears to lack a prominent claw tooth, unlike all three bristly species, with *K. hirsuta* intermediate [23–27]. If claw dentition in *K. puravida* is sexually dimorphic, it could suggest that different constraints on cheliped function between *K. puravida* and *K. tyleri* determines the evolutionary avenue for the development of male-male agonistic weaponry, or that the nature of male-male interactions are fundamentally different (e.g. pushing and sparring vs. puncturing) owing to other ecological factors (e.g. population density). One way to explore this further would be to see if *K. puravida* males exhibit more puncture wounds than females alongside analyses of footage to better characterise agonistic interactions in both species.

Based on the results presented in this study, it is unclear whether–or in what way–cheliped sexual dimorphism is prevalent in the other kiwaids. Given the elongated chelipeds of *K. hirsuta*, *K. gemma* and *K. araonae*, possibly indicating their primary use as platforms for microbial farming, cheliped sexual dimorphism throughout the size distribution may be absent or limited to more subtle features like claw shape and tooth architecture. Alternatively, *K. puravida* may be the exception, inhabiting low-flow sites around methane seeps where reduced microbial productivity necessitates prioritising waving behaviour over agonistic behaviour, constraining the potential evolutionary scope for cheliped sexual dimorphism.

This study attempts to make inferences about the feeding and reproductive ecology of yeti crabs based primarily on morphology. However, these inferences would need to be confirmed by either prolonged *in situ* observation in the deep sea and subsequent specimen collection to confirm sex (a challenge owing to the remoteness of deep-sea chemosynthetic habitats), or through observation in aquaria, which may be feasible in *K. puravida* where some individuals have been kept alive for several months in aquaria [44], but has yet to be demonstrated with *K. tyleri*, or other yet crabs which inhabit far greater depths. (> 1800 m). The sampling of *K. tyleri* in this study was non-random, precluding some population inferences about size distribution. In the future a random sampling approach might better determine if the largest individuals are indeed male as current and previous studies suggest for these species (and other kiwaids), although there are ethical considerations with over sampling species endemic to isolated locations. The question of whether *K. tyleri* or *K. puravida* represents the more basal or derived state for cheliped dimorphism can only be determined by the additional sampling of the other kiwaid species, including *K. hirsuta* where only one specimen was ever sampled [23]. However, opportunities for sampling kiwaids are extremely limited as mid-ocean ridges are very remote. For example, as far as we are aware, the hydrothermal vents on the Pacific-Antarctic Ridge in the SE Pacific have not been sampled since 2005 and vents on the East Scotia Ridge in the Southern Ocean since 2012.

## Conclusion

This is the first morphological study to investigate claw symmetry in Kiwaidae and claw allometry and sexual dimorphism in *K. tyleri*. Left and right claws are similarly sized in both males and females, but with a slight but significant tendency to left claw dominance, suggesting that claws are unlikely to be specialised for crushing and cutting as is the case for durophagous crabs, but may signify some task specialisation or may be a vestigial ancestral trait. While claw size differences between male and female yeti crabs have been previously reported, the results for *K. tyleri* presented in this study are the first to show differences in claw allometry between male and female kiwaids, indicative of either sexual selection for larger claws by females, male-male competition for reproductive access to females, or differential energy allocation (growth vs reproduction) between the sexes, or a combination of these. Conversely *K. puravida* individuals do not display this form of dimorphism, reflecting either differences in mating system, in

the nature and intensity of male-male competition, and/or in the functional use of the chelipeds between the two species. The extent of sexual dimorphism in the other kiwaid species is unclear, but this study represents an early step in elucidating the basic ecology of the yeti crabs, which is critical if highly endemic, geographically isolated species such as these are to be protected from present and future anthropogenic threats.

## Supporting information

**S1 Table. Measurements (mm) for *Kiwa tyleri* structures in this study to nearest 0.1 mm.**
CL = carapace length. PL = cheliped propodus length. PH = cheliped propodus height.
M = male. F = female.
(DOCX)

**S2 Table. Standardised major axis (SMA) regression model output showing the relationship between log-transformed propodus length/height and carapace length for male and female *Kiwa tyleri* and *Kiwa puravida* specimens as performed by the smatr R package.** Models initially compared slope angle between males and females and if non-significant, a comparison with slope elevation (y-axis offset) assuming equal slopes was made. 95% CI = 95% confidence interval, $R^2$ = regression model coefficient of determination, P1 = p-value for regression models against null hypothesis that variables are uncorrelated, Null = null hypothesis of test comparing male and female slopes angle/elevation, P2 = p-value, indicating significance of test with values < 0.05 highlighted in bold.
(DOCX)

## Acknowledgments

We wish to thank the crew and scientists aboard the two *RRS James Cook* expeditions (JC042, JC080), for the collection of the *Kiwa tyleri* specimens and Marc Martin and fellow technicians at the Institute of Marine Science in Portsmouth, UK for invaluable support in the lab. This study would also not have been possible without the initial work by Dr Sven Thatje and Vittoria Francis. Additionally, we thank the authors of Azofeifa-Solana *et al.* [44] for publishing their raw data which was invaluable for this study.

## Author Contributions

**Conceptualization:** Christopher Nicolai Roterman, Leigh Marsh, Jon T. Copley.

**Data curation:** Christopher Nicolai Roterman, Molly McArthur, Cecilia Laverty Baralle.

**Formal analysis:** Christopher Nicolai Roterman, Molly McArthur, Cecilia Laverty Baralle.

**Funding acquisition:** Jon T. Copley.

**Investigation:** Christopher Nicolai Roterman.

**Methodology:** Christopher Nicolai Roterman, Molly McArthur, Cecilia Laverty Baralle, Leigh Marsh, Jon T. Copley.

**Project administration:** Christopher Nicolai Roterman, Jon T. Copley.

**Supervision:** Christopher Nicolai Roterman.

**Validation:** Christopher Nicolai Roterman.

**Visualization:** Christopher Nicolai Roterman.

**Writing – original draft:** Christopher Nicolai Roterman, Jon T. Copley.

**Writing – review & editing:** Christopher Nicolai Roterman, Molly McArthur, Cecilia Laverty Baralle, Leigh Marsh, Jon T. Copley.

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
