## [Decision Letter · Decision Letter 0]

10 Oct 2024

PONE-D-24-41657Fighting for Favour: Sexual Dimorphism and Symmetry in the Hydrothermal Vent and Methane Seep Endemic Yeti Crabs (Kiwaidae).PLOS ONE

Dear Dr. Christopher Nicolai Roterman,

Thank you for submitting your manuscript to PLOS ONE. After careful consideration, we feel that it has merit but does not fully meet PLOS ONE’s publication criteria as it currently stands. Therefore, we invite you to submit a revised version of the manuscript that addresses the points raised during the review process.

 Overall the manuscript is  well written. However,  some major issues only with the methodology which with appropriate changes I think the manuscript could be accepted.

We look forward to receiving your revised manuscript.

Kind regards,

Murtada D. Naser

Academic Editor

PLOS ONE

Journal Requirements:

2. We noted in your submission details that a portion of your manuscript may have been presented or published elsewhere. [One component of the research was originally published in another research paper by Azofeifa-Solano et al. 2022. In our study we have re-analysed the data they published using an updated methodology.] Please clarify whether this [conference proceeding or publication] was peer-reviewed and formally published. If this work was previously peer-reviewed and published, in the cover letter please provide the reason that this work does not constitute dual publication and should be included in the current manuscript.

3. We note that Figure 4 in your submission contain [map/satellite] images which may be copyrighted. All PLOS content is published under the Creative Commons Attribution License (CC BY 4.0), which means that the manuscript, images, and Supporting Information files will be freely available online, and any third party is permitted to access, download, copy, distribute, and use these materials in any way, even commercially, with proper attribution. For these reasons, we cannot publish previously copyrighted maps or satellite images created using proprietary data, such as Google software (Google Maps, Street View, and Earth). For more information, see our copyright guidelines: http://journals.plos.org/plosone/s/licenses-and-copyright.

a. You may seek permission from the original copyright holder of Figure 4 to publish the content specifically under the CC BY 4.0 license. 

b.) If you are unable to obtain permission from the original copyright holder to publish these figures under the CC BY 4.0 license or if the copyright holder’s requirements are incompatible with the CC BY 4.0 license, please either i) remove the figure or ii) supply a replacement figure that complies with the CC BY 4.0 license. Please check copyright information on all replacement figures and update the figure caption with source information. If applicable, please specify in the figure caption text when a figure is similar but not identical to the original image and is therefore for illustrative purposes only.

Reviewers' comments:

Reviewer's Responses to Questions

**Comments to the Author**

1. Is the manuscript technically sound, and do the data support the conclusions?

Reviewer #1: Yes

Reviewer #2: Yes

Reviewer #3: Yes

2. Has the statistical analysis been performed appropriately and rigorously? 

Reviewer #1: No

Reviewer #2: No

Reviewer #3: Yes

3. Have the authors made all data underlying the findings in their manuscript fully available?

Reviewer #1: Yes

Reviewer #2: Yes

Reviewer #3: Yes

4. Is the manuscript presented in an intelligible fashion and written in standard English?

Reviewer #1: Yes

Reviewer #2: Yes

Reviewer #3: Yes

5. Review Comments to the Author

Reviewer #1: The present study is the first to show sexual dimorphism and symmetry of chelipeds in the Yeti crab Kiwa tyleri, endemic to hydrothermal vents and methane seeps. In addition, the authors re-analysed the allometric growth of chelipeds in another Yeti crab, Kiwa puravida, from published data and discussed interspecific differences in cheliped morphometry and functions. The manuscript is well written and will add new biological information on deep-sea Yeti crabs. Please consider the following minor comments.

Title: Please include the term "cheliped" or "claw" in the title.

L81-82. Please provide the scientific name for each Yeti crab photo.

L195. Does this require Bonferroni correction? Test for asymmetry index between sexes and laterality of chelae are independent.

L211-212. Again, is Bonferroni correction needed? You can use the 10 models shown in Table 2, but you estimated 40 coefficients including the intercept for 10 models. If you use the Bonferroni correction, you should adjust the p-values taking into account the number of parameter estimates. I do not think the Bonferroni correction is necessary for allometric growth analyses.

Reviewer #2: The manuscript in general is well written and brings important information on yeti crabs’ biology. I have some major issues with the methodology. The rest of the manuscript I have only minor suggestions.

Title: suppress “Fighting for favour” because there’s not enough evidence the all the analyzed species actually fight.

Introduction:

The introduction section is well constructed around these topics, however, to ensure a better readability I suggest removing these topics headings to create a continuous coherent text.

Line 60: remove “chirostyloid” or add somewhere before that these crabs belong to the Chirostyloidea superfamily.

Line 66: Figure1 � I suggest using only images of the species used the manuscript, and, if possible, pictures taken by the authors and not from some other publications.

Line 68: What SWIR means ? It is some sort of indication of the undescribed species mentioned earlier ? Clarify it.

Line 71: Remove figure 2 from the manuscript.

M&M

Line 152: JC080 – 02 December 2012 to 30 December 2010 ? Is this correct ?

Line 155: I guess here is Figure 4 ?

Line 163: Please explain “Kiwaids were not systematically or randomly sampled”. It is random or nor ? If it is random than they are not systematically sampled.

Line 173: Measurements:

Why did the authors took those measurements ?

The propodus length/height is understandable, but, the authors should explain why it was chosen.However, the carpus is not widely used in relative growth analysis. Thus, the authors should explain why the carpus was measured. The carpus has any biological/behavioral significance for the crabs ?

Line 175: Change “body parts” to “structures”. Change this throughout the text.

Line 181: distil ? or distal ?

Lines 200 – 212: The relative growth analysis here proposed are quite confusing. It is widely known that chelae x carapace relationships (and others) in crustaceans are not linear. In fact, it often assume an power function distribution and that’s why it is logarithmic transformed in these kind of studies.

Please see:

Huxley, J. S. (1950). Relative Growth and Form Transformation. Royal Society of London, 137(889), 465–469.

Huxley, J. S., & Teissier, G. (1936). Terminology of Relative Growth. Nature.

Hartnoll, R. G. G. (1978). The Determination of Relative Growth in Crustacea. Crustaceana, 34(3), 281–293.

Hartnoll, R. G. (1974). Variation in Growth Pattern between Some Secondary Sexual Characters in Crabs ( Decapoda Brachyura ). Crustaceana, 27(2), 131–136.

Hartnoll, R. G. (2001). Growth in Crustacea – twenty years on. Hydrobiologia, 449, 111–122.

These studies should provide a comprehensive knowledge on basic relationships in crustaceans’ relative growth.

Thus, using linear models (e.g linear regression) is a correct choice of analysis, however, there is some catch here. For bivariate relative growth, the residuals are important and a simple linear regression might sometimes provide a not accurate model. Thus, my suggestion is to change the analysis to the type II linear regression (major axis analysis) which are more accurate. This can be easily performed using the SMATr package built for R.

For a comprehensive discussion on bivariate allometric fitting lines please read:

Warton, D. I., Wright, I. J., Falster, D. S., & Westoby, M. (2006). Bivariate line-fitting methods for allometry. Biological Reviews of the Cambridge Philosophical Society, 81(2), 259–291. https://doi.org/10.1017/S1464793106007007

Warton, D. I., Duursma, R. A., Falster, D. S., & Taskinen, S. (2012). smatr 3- an R package for estimation and inference about allometric lines. Methods in Ecology and Evolution, 3(2), 257–259. https://doi.org/10.1111/j.2041-210X.2011.00153.x

Lines 213-217: During the re-analysis of K. puravida, it was not considered the carpus measures ? Why ? Then, why did the authors chose carpus measures for K. tyleri ?

Lines 217-220: The thankful note should be on the acknowledgement section.

Results

Lines 225-226: add the standard deviation with the mean values.

Lines 227 – 228: Why mention that the 10 largest specimens were male ? Remove.

I suggest to add some sort of histogram by size classes for males and females to better represent and visualize their size distribution.

Line 233: “For the propodus asymmetry index (AI), negative values indicate a left skew and positive valuesa right skew.” This information should be at the M&M section.

Lines 252 – 275: See the comments for methodology on relative growth/allometry.

Discussion:

There’s a lack of discussion around the carpus relative growth which is explained by the lack of explanation of is use/importance during introduction and methodology sections. Add a discussion on carpus or remove it from the manuscript.

Line 323: The authors suggest that K. tyleri may not employ carnivory, however, during introduction authors said that there are teeth and such on these crabs’ claws which suggested predatory behavior. Thus, the sentence here seems inconsistent. I believe that claws functions to push/pull and squeeze, thus the symmetry may not imply that they are not predators. If their prey is not hard/complex to acquire, thus, they don’t necessarily need a dimorphic high specialized claw. I suggest rethinking this sentence.

Line 355: “difference may reflect sexual selection relating to male-male competition for females”. Since these animals are blind, doesn’t seem feasible a male-male competition for female since size may not matter if there is no visual signal, although, some experiments could be done in this matter, which I personally think would be great! Thus, this difference may be related to differential energy usage. Males are often found close to the vents which in turn may have access to more/better food, whilst females are found marginally. Also, males tend to allocate more energy towards growth than females. Females also use a lot of energy each reproductive cycle, directing its energy towards egg production and so on, which can hamper the chelipeds’ growth. Therefore, in this sentence, I suggest a discussion towards different energy allocation between sexes.

Reviewer #3: The manuscript entitled "Fighting for Favour: Sexual Dimorphism and Symmetry in the Hydrothermal Vent and Methane Seep Endemic Yeti Crabs (Kiwaidae)" presents very robust and novel data for science. In addition, it uses data from a congeneric species for comparisons between these organisms, which deepened some discussions about the results.

In my opinion, the article is very well written and deserves to be published. I recommend a minor review due to a few sections where I believe the text needs corrections or further discussion. The text has a deficiency related to meeting basic zoological nomenclature rules. There are several instances where species are mentioned without proper citation of scientific authorities. Moreover, throughout the text, species names are written in full, whereas they should be abbreviated after the first mention. Although these errors are consistent and repetitive, they are easy to resolve after a careful review by the authors. Once these issues are addressed, I believe the manuscript can be accepted.

6. PLOS authors have the option to publish the peer review history of their article (what does this mean?). If published, this will include your full peer review and any attached files.

Reviewer #1: No

Reviewer #2: No

Reviewer #3: No

---

## [Author Response · Author response to Decision Letter 0]

4 Nov 2024

To the editorial office of PLOS One,

Response to editors:

Figure 4 showing a map has been removed as we feel it doesn't add to the manuscript, thus avoiding any copyright issues. Figure 2 was removed at the request of reviewer 2 for being superfluous. Figure 8 was removed as it reported cheliped carpus allometry, which reviewer 2 felt did not add anything to the study as this cheliped segment is not routinely measured when exploring claw sexual dimorphism in decapods.

Additionally, we wish to clarify that while the raw data for the K. puravida measurements included in this study for comparative purposes were originally published in the peer-reviewed study by Azofeifa-Solano et al., their analyses and conclusions were deeply flawed, and in our opinion should not have passed peer review. Consequently, we have re-analysed their raw data with an appropriate methodology (now updated as per reviewer comments) for comparison with the K. tyleri data generated in our manuscript. We do not consider this a duplication of previously published material, but an entirely new analysis.

Response to reviewers:

Please find below a response to each reviewer comment in turn. Additionally in the process of making the adjustments requested, there have also been some additional minor adjustments: including a more consistent usage of terms such as cheliped, chela and claw, the restructuring of some sentences for better clarity, the addition of some descriptive statistics relating to asymmetry (% left-right claw size discrepancy) and the reporting of an additional statistical test to detect asymmetry index deviations from 0 in Kiwa tyleri.

Reviewer 1 comments:

1) Title: Please include the term "cheliped" or "claw" in the title. 

• Amended. We have changed the title to include these terms.

2) L81-82. Please provide the scientific name for each Yeti crab photo. 

• Amended.

3) L195. Does this require Bonferroni correction? Test for asymmetry index between sexes and laterality of chelae are independent.

• Amended – Bonferroni correction removed. Manuscript adjusted as subtle left dominance is now significant.

4) L211-212. Again, is Bonferroni correction needed? You can use the 10 models shown in Table 2, but you estimated 40 coefficients including the intercept for 10 models. If you use the Bonferroni correction, you should adjust the p-values taking into account the number of parameter estimates. I do not think the Bonferroni correction is necessary for allometric growth analyses.

• Amended – Bonferroni correction removed.

Reviewer 2 comments:

1) Title: suppress “Fighting for favour” because there’s not enough evidence the all the analyzed species actually fight.

• Amended – title changed to remove reference to fighting.

2) The introduction section is well constructed around these topics, however, to ensure a better readability I suggest removing these topics headings to create a continuous coherent text.

• Amended – subheadings removed.

3) Line 60: remove “chirostyloid” or add somewhere before that these crabs belong to the Chirostyloidea superfamily.

• Amended.

4) Line 66: Figure1 I suggest using only images of the species used the manuscript, and, if possible, pictures taken by the authors and not from some other publications.

• We feel that a figure illustrating the bauplan of all kiwaid species has utility regarding discussion of the potential ecological and behavioural differences between species and the impact on claw sexual dimorphism. Original images of all species cannot be taken by authors as specimens are limited and distributed around the globe.

5) Line 68: What SWIR means ? It is some sort of indication of the undescribed species mentioned earlier ? Clarify it.

• Amended. This has been clarified to refer to Kiwa specimens from the Southwest Indian Ridge.

6) Line 71: Remove figure 2 from the manuscript.

• Amended. Figure two removed.

7) Line 152: JC080 – 02 December 2012 to 30 December 2010 ? Is this correct ?

• Amended.

8) Line 155: I guess here is Figure 4 ?

• Amended. Figure 4 has been removed. Text also amended.

9) Please explain “Kiwaids were not systematically or randomly sampled”. It is random or nor ? If it is random than they are not systematically sampled.

• Amended. The K. tyleri specimens were non-randomly sampled - as specific size cohorts were targeted.

10) Line 173: Measurements: Why did the authors took those measurements ? The propodus length/height is understandable, but, the authors should explain why it was chosen.However, the carpus is not widely used in relative growth analysis. Thus, the authors should explain why the carpus was measured. The carpus has any biological/behavioral significance for the crabs ?

• Amended. Propodus height and length taken for comparative purposes. All carpus measurements and reference to those measurements have been removed from the manuscript (with tables modified and figure 8 removed) as they essentially show the same pattern with propodus measurements, and cannot be compared with other studies in the literature. Figure

11) Line 175: Change “body parts” to “structures”. Change this throughout the text.

• Amended.

12) Line 181: distil ? or distal ?

• Amended – all switched to distal.

13) Lines 200 – 212: The relative growth analysis here proposed are quite confusing. It is widely known that chelae x carapace relationships (and others) in crustaceans are not linear. In fact, it often assume an power function distribution and that’s why it is logarithmic transformed in these kind of studies. 

Please see: Huxley, J. S. (1950). Relative Growth and Form Transformation. Royal Society of London, 137(889), 465–469. Huxley, J. S., & Teissier, G. (1936). Terminology of Relative Growth. Nature. Hartnoll, R. G. G. (1978). The Determination of Relative Growth in Crustacea. Crustaceana, 34(3), 281–293. Hartnoll, R. G. (1974). Variation in Growth Pattern between Some Secondary Sexual Characters in Crabs ( Decapoda Brachyura ). Crustaceana, 27(2), 131–136. Hartnoll, R. G. (2001). Growth in Crustacea – twenty years on. Hydrobiologia, 449, 111–122. 

These studies should provide a comprehensive knowledge on basic relationships in crustaceans’ relative growth. Thus, using linear models (e.g linear regression) is a correct choice of analysis, however, there is some catch here. For bivariate relative growth, the residuals are important and a simple linear regression might sometimes provide a not accurate model. Thus, my suggestion is to change the analysis to the type II linear regression (major axis analysis) which are more accurate. This can be easily performed using the SMATr package built for R. 

For a comprehensive discussion on bivariate allometric fitting lines please read: Warton, D. I., Wright, I. J., Falster, D. S., & Westoby, M. (2006). Bivariate line-fitting methods for allometry. Biological Reviews of the Cambridge Philosophical Society, 81(2), 259–291. https://doi.org/10.1017/S1464793106007007 Warton, D. I., Duursma, R. A., Falster, D. S., & Taskinen, S. (2012). smatr 3- an R package for estimation and inference about allometric lines. Methods in Ecology and Evolution, 3(2), 257–259. https://doi.org/10.1111/j.2041-210X.2011.00153.x

• Amended. Text has been modified to acknowledge that log-transformation is the default for allometric analyses, with convoluted parts of the text simplified and shortened. All allometric data have been re-analysed using type II regressions with the SMATr package and plots amended accordingly. Suggested references to Hartnoll et al and Warton et al papers have also been included.

14) Lines 213-217: During the re-analysis of K. puravida, it was not considered the carpus measures ? Why ? Then, why did the authors chose carpus measures for K. tyleri ?

• Amended. As per recommendations of Reviewer 1, cheliped carpus measurements have been removed as Azofeifa-Solano et al. did not take those measurements with K. puravida and the carpus measurements taken by us show similar allometric patterns to the propodus measurements.

15) Lines 217-220: The thankful note should be on the acknowledgement section.

• Amended.

16) Lines 225-226: add the standard deviation with the mean values.

• Amended.

17) Lines 227 – 228: Why mention that the 10 largest specimens were male ? Remove. I suggest to add some sort of histogram by size classes for males and females to better represent and visualize their size distribution.

• Partially amended. Mention of 10 largest specimens being male has been removed. We have chosen not to include a histogram, as samples were non-randomly collected, negating useful inferences regarding population size structure.

18) Line 233: “For the propodus asymmetry index (AI), negative values indicate a left skew and positive valuesa right skew.” This information should be at the M&M section.

• Amended. This information has been moved to the Methods section.

19) Lines 252 – 275: See the comments for methodology on relative growth/allometry.

• Amended.

20) There’s a lack of discussion around the carpus relative growth which is explained by the lack of explanation of is use/importance during introduction and methodology sections. Add a discussion on carpus or remove it from the manuscript.

• Amended. All mention of carpus measurements have been removed from the manuscript.

21) Line 323: The authors suggest that K. tyleri may not employ carnivory, however, during introduction authors said that there are teeth and such on these crabs’ claws which suggested predatory behavior. Thus, the sentence here seems inconsistent. I believe that claws functions to push/pull and squeeze, thus the symmetry may not imply that they are not predators. If their prey is not hard/complex to acquire, thus, they don’t necessarily need a dimorphic high specialized claw. I suggest rethinking this sentence.

• Partially amended. While several species of Kiwaidae do appear to have teeth on their claws, K. tyleri does not appear to have teeth. As such, this difference is discussed with the possibility that the lack of teeth on K. tyleri claws signifies that they use their claws in a different way and that durophagy is less likely. We have amended the text to acknowledge that K. tyleri may use their claws for manipulating/crushing/cutting soft food sources.

22) Line 355: “difference may reflect sexual selection relating to male-male competition for females”. Since these animals are blind, doesn’t seem feasible a male-male competition for female since size may not matter if there is no visual signal, although, some experiments could be done in this matter, which I personally think would be great! Thus, this difference may be related to differential energy usage. Males are often found close to the vents which in turn may have access to more/better food, whilst females are found marginally. Also, males tend to allocate more energy towards growth than females. Females also use a lot of energy each reproductive cycle, directing its energy towards egg production and so on, which can hamper the chelipeds’ growth. Therefore, in this sentence, I suggest a discussion towards different energy allocation between sexes.

• Amended. In the discussion, a distinction has been made between differential energy allocation between males and females, male-male competition for access to females and sexual selection. All three, either separately or in combination are acknowledged as explaining claw sexual dimorphism. The possibility of sexual selection has been mooted via tactile signalling, rather than visual signalling.

Reviewer 3 comments:

1) The text has a deficiency related to meeting basic zoological nomenclature rules. There are several instances where species are mentioned without proper citation of scientific authorities. Moreover, throughout the text, species names are written in full, whereas they should be abbreviated after the first mention.

• Amended. Species names have been abbreviated as per the reviewer's instructions. In the first instance of a species name, the authority is cited.

2) (In-text comment, line 41) I recommend the removal of all subtitles in this section. This is not required by the journal’s guidelines, and, moreover, it is uncommon and results in excessive text breaks. Keep a standard structure for a scientific article: Introduction, Material and Methods, Results, Discussion, and Conclusions. Subtitles are more commonly used in Material and Methods and Results, as authors can implement different types of analyses for each type of result.

• Amended. Subtitles removed.

3) (In-text comment, line 81) Your figure includes letters that likely represent different species of the genus Kiwa. However, I cannot be certain of this because you did not provide this information in the figure legend. You should identify what each letter represents in the legend, from A to F.

• Amended. Species identified in Fig 1 caption.

4) (In-text comment, line 100) I do not think this can be stated. Because the cheliped is a modified pereiopod, it does not have a locomotory function, so we cannot refer to it in that way.Therefore, just mention that chelipeds are the first pair of pereiopods (for most decapod crustaceans, with some exceptions).

• Amended. Walking appendage has been changed to pereopod.

5) (In-text comment, line174) Include a citation of the study you used as the basis for this verification.

• Amended. Thatje et al. 2015 referenced for sexing of females.

6) (In-text comment, line 334) This discussion is interesting. However, symmetry is also a characteristic observed in other Anomura, such as some representatives of the family Porcellanidae, which also inhabit corals. In this family, there are several symmetric and asymmetric groups. If you would like to extend the discussion to this family, I believe it would be relevant.

• Amended. Although conclusions about symmetry have now been modified to acknowledge the subtle but significant left claw dominance in K. tyleri, the discussion about claw symmetry and asymmetry has been widened not only to include Porcellanidae, but also Aegloidea, and Munididae within Anomura.

We hope that the extensive amendments made to the manuscript addresses the stylistic, methodological, analytical and discussion concerns of the reviewers.

---

## [Editor Report · Decision Letter 1]

8 Nov 2024

Yeti Claws: Cheliped Sexual Dimorphism and Symmetry in Deep-Sea Yeti Crabs (Kiwaidae).

PONE-D-24-41657R1

Dear Dr. Christopher Nicolai Roterman,

We’re pleased to inform you that your manuscript has been judged scientifically suitable for publication and will be formally accepted for publication once it meets all outstanding technical requirements.

Kind regards,

Murtada D. Naser

Academic Editor

PLOS ONE
---

## [Editor Report · Acceptance letter]

17 Nov 2024

PONE-D-24-41657R1 

PLOS ONE

Dear Dr. Roterman, 

I'm pleased to inform you that your manuscript has been deemed suitable for publication in PLOS ONE. Congratulations! Your manuscript is now being handed over to our production team.

Kind regards, 

on behalf of

Dr. Murtada D. Naser 

Academic Editor

PLOS ONE